# Electrical Bioimpedance Analysis for Evaluating the Effect of Pelotherapy on the Human Skin: Methodology and Experiments

**DOI:** 10.3390/s23094251

**Published:** 2023-04-25

**Authors:** Margus Metshein, Varje-Riin Tuulik, Viiu Tuulik, Monika Kumm, Mart Min, Paul Annus

**Affiliations:** 1Thomas Johann Seebeck Department of Electronics, Tallinn University of Technology, Ehitajate tee 5, 19086 Tallinn, Estonia; 2West Tallinn Central Hospital, Paldiski Mnt. 68, 10617 Tallinn, Estonia; 3The Centre of Excellence in Health Promotion and Rehabilitation, Lihula Mnt. 12/1, 90507 Haapsalu, Estonia; 4Pärnu College, University of Tartu, Ringi 35, 80012 Pärnu, Estonia

**Keywords:** curative mud, electrical bioimpedance, electrode, impedance spectroscopy, mud pack, pelotherapy, skin barrier

## Abstract

Background: Pelotherapy is the traditional procedure of applying curative muds on the skin’s surface—shown to have a positive effect on the human body and cure illnesses. The effect of pelotherapy is complex, functioning through several mechanisms, and depends on the skin’s functional condition. The current research objective was to develop a methodology and electrodes to assess the passage of the chemical and biologically active compounds of curative mud through human skin by performing electrical bioimpedance (EBI) analysis. Methods: The methodology included local area mud pack and simultaneous tap water compress application on the forearms with the comparison to the measurements of the dry skin. A custom-designed small-area gold-plated electrode on a rigid printed circuit board, in a tetrapolar configuration, was designed. A pilot study experiment with ten volunteers was performed. Results: Our results indicated the presence of an effect of pelotherapy, manifested by the varying electrical properties of the skin. Distinguishable difference in the measured real part of impedance (*R*) emerged, showing a very strong correlation between the dry and tap-water-treated skin (r = 0.941), while a poor correlation between the dry and mud-pack-treated skin (r = 0.166) appeared. The findings emerged exclusively in the frequency interval of 10 kHz …1 MHz and only for *R*. Conclusions: EBI provides a promising tool for monitoring the variations in the electrical properties of the skin, including the skin barrier. We foresee developing smart devices for promoting the exploitation of spa therapies.

## 1. Introduction

Mud is a blend of organic and inorganic substances dissolved in water that has experienced different geological and biological processes in a natural (or artificial) physiochemical environment [1]. Natural thermal muds are traditionally used in therapeutic procedures. However, presently, cosmetics are typically processed (ground and mixed with other substances such as peat or clay). The processed thermal muds are used for therapeutic purposes under the name of peloids [2].

Peloids are aimed at alleviating different ailments in traditional medicine and for application onto the skin in cosmetology, called pelotherapy under the general term balneotherapy [2,3]. Balneology is a research field of baths and bathing in natural waters for healing purposes, including the application of the emerging gasses and peloids [4]. A method of pelotherapy is mud pack compress application—an option treating localized areas of the body to increase skin permeability and microcirculation [5] or reducing pain in the case of knee osteoarthritis [6]. The popularity of balneotherapy is increasing [7], related to the perpetually-appearing new evidence of the advantageous effects of peloids on the human body [8] and awareness of personal health. The utilization of peloids in natural cosmetics is also a rising trend, appearing in the wellness and relaxation field of medical and cosmetological applications [9].

It is generally accepted that the skin, which is a very complex structure, is affected by a set of different factors of pelotherapy. Based on scientific evidence, the pelotherapeutic procedures are proposed, e.g., to relieve the joint pain of osteoarthritis of the knees [10], to have a positive effect on gynecological problems [11], to entail the passage of minerals through psoriatic skin [12], and to enhance the superficial blood circulation [5]. Pelotherapy has a useful effect on muscle tone and reduces pain through increasing temperature and hydrostatic pressure [13].

The effect of pelotherapy can be evaluated directly or indirectly, from which the latter includes questionnaires and visual examination. The direct methods can be divided based on the utilized techniques, for example optical (laser Doppler flowmetry) [5]; electrical (corneometry—for estimating the relative permittivity of the skin [14]; electrical bioimpedance (EBI)); and blood analysis. A comprehensive overview of skin assessment techniques is given elsewhere [15].

The primary front for peloids in balneotherapy is the skin—the largest organ in the human body, which is connected to the nervous system. Roughly, the stimulation of the skin by hot mud during pelotherapy is transferred through the sympathetic nervous system to the organs, which take part in metabolic processes of the human body [3].

The human skin is a layered structure of different cell types, which perform various functions. Typically, the three layers of the skin are denominated: epidermis, dermis, and hypodermis. The epidermis can be divided into sublayers, starting with the outermost layer: stratum corneum (SC), stratum granulosum, stratum spinosum, and stratum basale [16]. The SC at the forearm is about 10–20 µm thick and composed of stratified layers of dead and flattened cells, i.e., corneocytes [17,18]. The SC can be depicted as a brick wall, consisting of corneocytes’ extracellular matrix [19]. The impedance of the SC depends on the moisture level in the air and sweating and can be considered as a dielectric in dry conditions [20]. The remaining layers of the epidermis consist of nucleated cells and are typically 50–100 µm thick [21].

The dermis of the forearm is roughly 1 mm thick and consists of elastic fibers, matrices of structural proteins, and cellulose—providing elasticity and strength to the skin. Together with the epidermis, the thickness has been reported to be 1.16–1.22 mm [22]. The electrical conductivity of the dermis is reported to be higher than for the epidermis [23]. The hypodermis is the bottom layer of the skin, consisting mainly of body fat together with blood vessels and nerves. The conductivity of the hypodermis has been reported to be low, however higher than that of the epidermis [23].

It has been found by Martinsen et al. [24] that, at low frequencies (below 1 kHz), the measured impedance of the skin is dominated by the SC. At higher frequencies (above 100 kHz), the viable skin dominates. Furthermore, the same authors showed that, at frequencies beyond 1 MHz, the contribution of the SC becomes very low, about 5% [24].

EBI measurements have been successfully utilized in cardiac pacemakers: the impedance of the cardiac muscle is monitored to evaluate the energy balance to automatically set the rhythm [25]. Furthermore, the dynamic change monitoring in the human body, caused by respiratory and cardiac activity, has been implemented—for example, to estimate the central aortic pressure of the blood [26]. However, the EBI analysis of the skin has been implemented with various goals—e.g., to monitor skin cancer [27], wound healing [28], and the effectivity and the result of transdermal drug delivery [29,30].

The effect of curative mud on the human body has previously been researched by several studies on large groups of volunteers. Using laser Doppler flowmetry, the improvement in blood supply in the skin after the spa therapy procedures was confirmed statistically—more specifically, in the pelotherapy group [31]. The same outcome was gained in a study including different blends of mud and peat, where the joint effect of both substances was determined: the moistening of the skin [32].

The development of personalized medicine and the revolution in wellness towards sustainable and environmentally friendly technologies introduce the actuality of the object of our ongoing research. The elaboration of a precise and versatile methodology for assessing the effect of cosmeceutical substances on the skin constitutes a decent base for the further development of natural compounds in skin care products. Balneology, in general, and, more precisely, pelotherapy, with its complex palette of compounds, the complex effect of which on the human skin is assumed to provide surprising results, is already gaining popularity. The importance of the development and improvement of the means for detecting and discovering its effect is evident.

In this paper, we present the results of a pilot study to assess the effect of pelotherapy on the human skin (and underlying tissues) by using EBI analysis. Such research, including the specific effect of mud therapy, comprising the ability and different mechanisms of different substances to penetrate the skin barrier, by means of EBI has not been reported before. We introduced a measurement methodology by using local area mud pack application and custom-designed small-area electrodes on rigid printed circuit boards for monitoring the EBI of the skin. We demonstrate our findings by the determination of the effect of mud in comparison with the effect of tap water and the dry status of the skin in the case of measurements on both forearms. The clear difference between the mud-pack-treated and tap-water-treated skin appeared in the real part of the impedance in the frequency interval of 10 kHz …1 MHz, comprising the contribution of the current research.

The main purpose of the performed experiments was to create a methodology and increase the knowledge of the advantageous effect of pelotherapy to promote the application of mud therapy. This research project was ultimately expected to establish novel solutions and instrumentation that alter balneotherapy procedures to the current smart wellness and medical Internet-of-Things-based automatic supervision. The outcome was expected to provide prompt feedback on the effect of the performed pelotherapy to doctors and patients.

## 2. Background

In this section, the background of the research is presented, introducing the basics of EBI measurements and the mechanisms of curative mud in contact with the skin.

### 2.1. Basics of EBI

Electrical impedance presents the ratio between the voltage and current and reveals in both cases: DC and AC. While in the case of DC, only the resistive component is applied, in the case of AC, the current flow is inhibited by the reactive component. This reactive component originates from the inductive and capacitive phenomena: the voltages that are induced in the matter by self-induced magnetic fields and the electrostatic phenomenon of charge storage by capacitive elements.

The impedance *Z* is a complex term that can be divided into real (resistance *R*) and imaginary parts (reactance *X*) as
(1)Z=ReZ+jImZ=R+jX.

Under linear conditions in the case of the same tissue, the unity cell inverse of *Z* (admittance *Y*) contains the same information as the unity cell Z—however, shown differently (for a full theoretical description, see [20]). *Y* can be calculated (where *G* is conductance and *B* is susceptance) as
(2)$$Y=Z^{-1}=ReY+jImY=G+jB.$$

The data can be presented also as the modulus and phase angle (θ). Yet, in the case of the human skin, the conditions are far from linear, i.e., the real and imaginary parts in both domains are expected to incorporate useful data. Moreover, it is known that the conductivity and permittivity of the biological object (including the skin) are frequency-dependent, i.e., the recognition of different dispersions is provided (firstly by Schwan in [33]). The appearance of different dispersions (α, β, δ, and γ) depends on the specific process or property of biological objects at different frequencies.

### 2.2. Effect of Curative Mud on the Skin

The effect of pelotherapy is claimed to be multimodal and can be divided into two domains: non-specific and specific [34]. The non-specific effect is a combination of several mechanisms: mechanical, thermal, sorptive, etc. [3,13]. The non-specific effect is claimed to be dependent on the functional condition of the vegetative nervous system and metabolism, which emerges as a counter-reaction to the accompanying mechanical and thermal effect of pelotherapy [34].

The specific effect originates from the influence of chemical substances on the skin’s surface and the action of the biologically active compartments of curative mud that soak into the human body. The detailed aspects of the specific effect of pelotherapy and the soaking of organic compounds and minerals through the skin are still debatable [35]. It is claimed that the chemical influence of peloids is revealed by the absorption of volatile substances: organic and sulfuric acids, salts, etc. [3]. The latter poses a question concerning the permeability of the skin to different substances.

The main front that attains physical contact with the mud is the skin. Based on the concept of the skin barrier, the skin has the protective function of a water–lipid jacket that is vastly associated with the epidermis—specifically with the SC [36]. The epidermis is a chemically active dynamic system that reacts to external factors by the synthesis of body cells, deoxyribonucleic acid, and ion transport [37]. In other words, skin is a selectively permeable barrier, which permits the permeation of different substances at different rates [16]. Thus, the chemical effect of mud therapy through the absorption of organic substances and minerals through the skin can theoretically be expected [13].

Generally, a molecule can penetrate the skin in two ways: through the epidermal pathway and the appendageal pathway. In the case of the epidermal pathway, the substance diffuses through the skin layers, doing this either transcellularly or intracellularly. In the case of the appendageal pathway, the sweat ducts and hair follicles provide gateways for molecules through the skin (from which the first one is denoted as predominant) [16]. Under the appendageal pathway, also pores must be noted [38].

The reduction of the skin barrier by using active methods is widely known, from which the simplest is tape stripping—the removal of the SC [39,40] (a comprehensive overview can be found from [41]). The most-used electrical methods for reducing the skin barrier are iontophoresis and electroporation. The first one includes the application of continuous low current excitation to drive macromolecules through the skin, while the second one incorporates the utilization of short high-voltage pulses to rearrange the SC lipids [41].

As the electrical treatment of skin for implementing transdermal drug delivery is gaining increasing attention in pharmaceutics, the interest in monitoring the amounts of delivered drugs is increasing as well, specifically, e.g., to discriminate the electrical effects of electrical treatment from the electrical characteristics of drug delivery in the skin [30]. EBI analysis has been successfully applied for measuring the characteristics of the skin after a treatment [29,42,43]. Nevertheless, the similarity between the delivery of drugs and the solution of organic substances or minerals through the skin can be recognized.

Here, the question concerning the layers of the skin appears: On which layer should the attention be concentrated to monitor the effect of curative mud? If the assumption is that the organic substances and minerals penetrate the skin during the mud therapy, attention may be turned to the epidermis as a layer associated primarily with a protective function. However, if the substances have passed the SC, they are expected to influence the conductivity of the dermis. In this sense, however, the expectation is that both the epidermis and dermis are influenced by pelotherapy.

The skin is also involved in thermoregulation, implemented partly through sweating. The sweat ducts (but also the hair follicles) constitute direct pathways through the whole layered skin structure. Due to its composition (the domination of sodium and, in a lower amount, potassium), sweat can be considered a good electrical conductor (with a reported median conductivity of 5.56 µS cm^−1^ in [44]). In dry conditions, the sweat ducts are empty or only partially filled. As a result, in the case of using standard pre-gelled ECG monitoring electrodes for measuring the EBI, the gel penetrates the ducts [45]. As the conductivity of gels and hydrogels of wet electrodes is typically much higher, the conductivity increases even more [46].

However, there is a lack of papers published on the topic of evaluating the effect of pelotherapy on human skin by using EBI analysis. The research is typically performed by using commercial devices by just reading and processing the displayed values in situ [6,47,48,49]. Impedance spectroscopy and analysis provide supplementary options for the characterization of the skin [50]. The modeling of impedance data [51] and fitting into an empirical model [52,53] serve as a potential tool for evaluating the skin and the effect of different environmental and medical (well-being) agents [30].

## 3. Materials and Methods

In this section, the requirements and the data of the volunteers are presented together with statistical analysis. The used materials (including the mud) with the description of the performed measurement procedures of EBI are described. Ultimately, the implemented method and developed EBI measurement electrodes are disclosed with insight into the model-based approach of representing the biological materials.

### 3.1. Choice and Data of Subjects

All the volunteers were introduced to the prepared information and agreement form and signed it thereupon. Secondly, a short health questionnaire was filled out by all the volunteers, requesting basic information concerning their health. After the review of the filled questionnaires by the doctors, ten healthy volunteers were chosen based on the following attributes:Adult;Absence of chronic diseases of the circulatory system, skin disorders, and acute illnesses.

Based on the filled forms, the data of the subjects were gathered, and the Body Mass Index (BMI) was calculated (the calculated mean values and standard deviations (σ) for the whole sample size can be seen in Table 1).

All of the volunteers were Caucasian and belonged to the interval of 20–40 years of age.

### 3.2. Materials

In this research, the focus was set on the determination of the effect of curative mud on human skin by electrical means. A mixture of wet unheated disintegrated mud of Haapsalu and peat was utilized. Based on the study of Estonian curative mud deposits by Terasmaa et al. [54], the mud of Tagalahe Bay in Haapsalu is largely minerals, containing on average 11.7% of organic material.

For the experiments, a bag of synthetic fabric (5 × 10 cm) was filled with a mixture of mud and peat (1 tablespoon) (mud compress). The synthetic fabric enabled a small amount of mixture to remain outside and cover the skin surface. To exclude the placebo effect and compare it with the impact of some other wet substance, a piece of the same fabric, wetted with tap water at room temperature, was applied on the other forearm simultaneously (water compress).

The compress together with the arm was covered by thin polyvinyl chloride (PVC) film to avoid mud and water drying during the application.

In the current research, tape stripping was not used to disrupt the skin barrier and ease the passage of substances. This decision was based on the fact that such a practice is not common in pelotherapy and mud pack therapy. Instead, our goal was to imitate the actual conditions of the targeted procedure.

### 3.3. Measurement Procedure

Four measurement cycles were performed in the cases of each volunteer:EBI measurement of the skin on the left forearm (A);EBI measurement of the skin on the right forearm (B);Application of the mud compress on the left forearm and the subsequent EBI measurement of the skin (C);Application of the water compress on the left forearm and the subsequent EBI measurement of the skin (D).

The choice of the forearms was intentional because the correlation between the BMI and the skin thickness in this body area has been reported to be the lowest [22]. Furthermore, the thickness of the SC at the ventral side of the forearm is reported to be one of the lowest in the human body [55], while diverging in the left and right forearm on average 0.03 mm [22]. There is a difference in the thickness of the skin based on sex. The average skin thickness of men on the ventral side of the forearm is 1.26 mm and, for women, 1.12 mm.

The procedure in the cases of Measurement Cycles A-B is described step-by-step in the following listing:The volunteer was sitting on the chair with the hand lying on the armrest at the height of about the last rib in the case of a horizontally bent arm (with the inner side of the forearm slightly exposed upwards);The area under the electrode on the surface of the skin was slightly moistened with a wet paper towel;The electrode was placed on about the area of the centerline of the middle side of the forearm and fixed by using a gentle medical tape (Figure 1);EBI measurement was performed 3 min after the electrode was attached to the skin surface;The tape with the electrode was removed.

The procedure in the cases of Measurement Cycles C-D is described step-by-step in the following listing:The compress of mud (Cycle C) (Figure 2a) or water (Cycle D) (Figure 2b) was placed on the inner side of the forearm and covered with PVC film.The PVC film and compress were removed after 30 min.Step 1 in the case of Measurement Cycles A-B.Loose water and mud were removed from the surface of the skin by using a paper towel.Step 3 in the case of Measurement Cycles A-B.Step 4 in the case of Measurement Cycles A-B.Step 5 in the case of Measurement Cycles A-B.

Three repetitive measurements were performed in each cycle.

### 3.4. Measurement Method

The impedance measurements were performed by using the four-electrode measurement system. The choice was made to avoid the electrode polarization impedances of current-carrying (CC) electrodes (applied in the case of two-electrode systems). An important aspect to consider hereby is the depiction of an object (in the current paper, the human skin) by its measured electrical behavior. In the case of a four-electrode system, the trans-impedance is a more precise term to use than impedance. In the case of a four-electrode system, two two-electrode systems are present—one for excitation and one for measurement [20].

Thus, the geometrical aspect must be considered due to the expected formulation of sensitivity fields in an object because of the possible interposition of voltage and current lead fields. Therefore, besides the zero sensitivity, positive and negative sensitivities may appear, affecting the total measured impedance [20].

The electrodes play a substantial role in the electrical measurement of biological objects, explained by the different types of current carriers when compared to solid matter. A comprehensive definition is provided in [20]: the electrode is the site of a shift from electronic to ionic conduction and vice versa. What happens is the exposure of layers of different materials, which, in the case of low-frequency measurements, gives birth to interfacial polarization [56]. This is caused by charge accumulation at the interface of the electrode and object (SC at non-invasive measurements) in the presence of the external electric field.

Moreover, in the case of non-invasive measurements, the electric connection strongly depends on the status of the object’s surface—the skin. These measurements incorporate time-dependent factors such as hydration, sweating activity, and emotional state [50]. However, the measured data must be gathered simultaneously, i.e., the signals must be processed and evaluated in correlation with the physical and emotional state of the subject and the surrounding environment.

In the case of electrolytes (i.e., wet SC at non-invasive measurements), the emergence of an electrical double-layer must be considered. The double-layer is caused by the contact of materials of different molecular structures where only the electrolytic side contributes (as the electrodes in metal are strictly bound) [20]. The electrical double-layer exhibits capacitive properties that are in series with the electrode; however, it is never a pure parallel-plate capacitor, as this layer is leaky and potential- and concentration-dependent [56].

The electrode material is a relevant issue as the polarization phenomenon is related to this choice. The non-polarizable electrodes are expected to achieve resistive contact with the object, while in the case of polarizable electrodes, the contact is expected to be capacitive. However, in practice, the electrodes are somewhat polarizable and somewhat non-polarizable, i.e., experiencing a finite faradaic impedance [57].

Other important aspects are the shape, dimensions, and distances between the electrodes when measuring the impedance of human skin. It has been reported that, in the case of a two-electrode system, the distance between the electrodes influences the measuring depth. Specifically, the penetration depth of currents in the object is half of the distance between the electrodes [20,27]. However, as discussed before, in the case of a four-electrode system, the contribution of different layers in the object is more complicated because of the geometrical aspect. On the other hand, the electrode area can be expected to influence the sensitivity in the gap between the electrodes; the narrower the gap, the higher the sensitivity is [20].

The importance of the choice of the shape and dimensions of the electrodes appears when measuring the human skin—related to its relatively small thickness and the presence of the sweat ducts. The gap between the electrodes cannot be smaller than the diameter of sweat duct (approximately 20 µm); otherwise, the small-sized electrodes could face a shunt path due to the relatively well-conducting sweat. This can be taken as a minimum gap; however, the conductivity of the SC is variable, and sweating SC can shunt the electrodes in applicable conditions.

Standard pre-gelled Ag/AgCl electrodes are very useful due to their standard design; however, problems appear. The electrode gel affects the SC through several mechanisms (for a comprehensive overview, see [46]). Importantly, the electrode gel may depend on the type or decrease or increase the electric parameters of the SC and influence the measured Y [20,46]. No less important, as expected in [20], the substances from the electrode gel may penetrate the SC, influencing the measured electrical characteristics of the skin [50].

Another possible way to approach the skin is to use interdigitated electrodes [58], which could be used to attain the mean of the underlying area on the skin. Another option is to use the quasi-monopolar configuration, which is reported to provide the feature of targeting the sensitivity of the skin area of interest [59].

For the current research, the choice was made to design gold-plated electrode contact surfaces on rigid FR-4 material (with a thickness of 1.6 mm) as a printed circuit board (PCB). The electrodes were manufactured by a standard rigid PCB manufacturing process: etching (1), photoengraving (2), and laminating (3) (no more additional steps were need as it was a simple two-sided PCB with no holes). On the top side of the PCB, copper fields for surface-mountable connectors were formed (Figure 3b). The copper fields on the electrode side (bottom) were galvanically gold-plated. The design is explained by the desire to ease the connection of the impedance spectroscope to the electrodes. The rigid electrodes provide the advantage of maintaining the fixed gap between the electrode contact surfaces and the possibility of attaching the designed PCB to the skin’s surface by using sticky tape. For the planned pilot study, the application of a miniature simplistic design of electrodes is valid as the effect of pelotherapy can be expected to apply equally on the skin surface.

The gap between the electrodes was 2 mm and the width of the electrode contact surface 1 mm (for the dimensions, refer to Figure 3a). The choice is explained by the wish to concentrate on a small skin area with the presumption of its relative uniformity. The electrode contact surfaces were galvanically gold-plated. The choice of gold plating originated from the expectation that the capacitive properties of the SC dominate in the dry condition of the skin [57]. As a result, the polarization properties of gold are hidden by the capacitive properties of the SC, which are of a much larger scale. However, with the increase of moisture in the SC and sweating, its capacitive properties disappear quickly and are replaced by resistive ones. However, gold is durable and has constant properties when compared to medical-grade gel electrodes, which are inconvenient in application.

### 3.5. Used Devices

For measuring EBI, a laboratory on-desk impedance spectroscope HF2IS together with an HF2TA transimpedance amplifier from Zurich Instruments was used. This device proposes the frequency range of 0.7 μHz–50 MHz and the possibility of using two- and four-electrode measurements [60].

The question of the excitation signal was solved by using a wide frequency sweep in the range of 100 Hz–20 MHz with an amplitude of 500 mV. However, during the data analysis, the suitable frequency intervals were sorted.

The question, accompanying the planning of electrical measurements of biological objects, is the choice of the parameters that would incorporate the data of interest. In the case of variable biological processes—such as the flow of pulsating blood in arteries—the ratio of change in the volume can be calculated and compared in situ while rejecting the uncertainties this originates from the interface between the skin and electrode [61]. However, when the focus is to measure the absolute value of *Z*, the object’s (i.e., the skin) properties must be considered thoroughly. For this reason, the real and imaginary parts of both—*Z* and *Y*—were gathered, compared, and evaluated.

As *Z* is a complex term (so is *Y*), consisting of the real and imaginary parts, originating from the layers of matter of better and worse conductivity, the consideration is the model-based representation of the skin.

### 3.6. Skin Impedance Models, i.e., the Method

The interfacial polarization problem can be addressed when depicting the structural composition of the skin. Biological objects can be represented by an equivalent circuit that consists of series and parallel combinations of resistors (*R*) and capacitors (*C*). These circuits behave very differently under the excitation of different frequencies. In the case of a series combination of RC circuits, the result is the summation of both elements. In the case of a parallel connection, the frequency dependence appears; at the low frequencies, the *Z* is contributed by resistors, while at the high frequencies, by capacitors [20].

Based on the above description, the electrical model of complex biological tissue (and skin) cannot be presented just by using series and parallel RC circuits. To add the frequency dependence, the inclusion of a constant phase element (CPE) is typically performed. The CPE depicts a non-ideal capacitor that is used instead of an ideal capacitor (α = 1) in the circuit together with a resistor (α = 0).

The impedance of the CPE ($Z_{CPE}$) can be rewritten as
(3)$$Z_{CPE}=1/{\left(j\omega C\right)}^{\alpha }.$$

The dependence of the impedance of an object to the frequency is described by Cole’s equation:(4)$$Z=R_{\infty }+\left(R_{0}-R_{\omega }/1+{\left(j\omega \tau_{0}\right)}^{\alpha }\right),$$

*Z_CPE_* is an empirical function, used in fitting the measured impedance of a biological object [62]. The parameter α denotes the displacement of the center of the circular arc below the real axis in the impedance phasor diagram. The Cole model is popular for characterizing the impedance data of biological objects because of its mathematical plainness and simple empirical equivalent circuit. Still, the utilization of RC layered models has been presented in the literature as well [58,63]—being in some cases complicated combinations of parallel and series RC circuits with more than 20 parameters [62].

In combining these two models, the most accurate one is expected to be achieved by adopting the *Z_CPE_* element in each layer’s RC structure [62]. However, we cannot describe the structure and dimensions of the skin layers of all subjects individually. However, generally, the structure can be described. A commonly accepted electrical model of the skin is a distillation of the previously noted model, consisting of the parallel connection of DC conductance and the polarization impedance of the system (both frequency0dependent) in series with resistance R∞, denoting the hypodermis [39,53,64].

The equation of *Z* being equal to the inverse of *Y* is true only in the case of linear conditions (homogeneous tissue structure). However, it does not apply to the real and imaginary parts of *Z* and *Y* because of the frequency dependence of the complex layered and inhomogeneous structure of the skin [20]. The data representation must be performed based on the chosen model.

Typically, *Z* is used to describe the object when it is represented as layers of different conductivity (parallel connections of single resistors *R* and single capacitors *C*) in series (Figure 4a), and *Y* is used when the layers are in parallel (Figure 4b) [20].

In the case of the parallel connection of the layers, the calculation of *Y* is straightforward: G1 + G2 and C1 + C2. In the case of the series model, the calculation gains a much more complex character because of the accompanying frequency dependence (at low frequencies, the current prefers to flow through the resistors and, at high frequencies, through the capacitors) [20].

For measuring the impedance of the skin, the series model is typically used (because of the layered structure of the skin); however, when considering the equivalent skin model, one can realize that it is not so straightforward. It can be imagined that the skin consists of several parallel and series RC connections, and the behavior is a combination of the response of these connections. Ultimately, this means that the data of interest are not expected to be incorporated only into the measured *Z* (series connection), but also into the measured *Y* (parallel connection).

Based on this consideration, the decision was made to gather altogether 3 datasets in each measurement cycle:*Z* and θ;*R* and *X*;*G* and *B*.

However, Reference [53] stated that, concerning skin hydration, all the necessary information can be found by implementing the low-frequency *B* measurements at a single frequency. This is explained by the influence of sweat ducts, which provide direct pathways through the skin while expectedly containing a minor amount of information about the skin (when measuring *G*). Still, the effect of curative mud on the skin is expected to be the result of a combination of different factors [13,34]. At the same time, this witnesses that the useful data may be contained by the resistive parts as well.

## 4. Measurement Results

The datasets from the impedance spectroscope were in the .csv format, while the processing and analysis were implemented in Microsoft Excel.

In the cases of the results of Cycles A and B, the dispersion was larger than in the cases of Measurement Cycles C and D. This was the result of uncertainties, caused by the interface between the dry electrode contact surfaces and the dry skin. The quality of the electrical connection in the described case depended on the level of sweating and the air humidity, i.e., dominated by the status of the SC. If the electrode is set on the skin for three minutes, the conditions equalize at some level; however, the dispersion of the results will still be larger than in the case of wetting the skin for 30 min with a wet compress (mud or water). The building up of water contact with the dry electrode may take 15 min or more [20].

To analyze and represent the measurement data in this section, the mean of three repetitive measurements was used.

The research was carried out by following the principles embodied in the Declaration of Helsinki and local statutory requirements. The gathered data were subjected to anonymization in all cases by adding an identification number to the subject; the gathered data cannot be connected to specific individuals. The study was performed under the approval of the Tallinn Medical Research Ethics Committee (Estonia) (Decision No. 2542). No difference between the sexes of the volunteers in the results was focused on nor noticed.

### 4.1. Verification of the Measurement Setup

To verify the chosen setup for the impedance measurements of the skin, an experiment in the cases of circuits of passive discrete electronic components was performed. The measurement was performed in the range of 1 kHz–2 MHz, as expectedly, at the lower frequencies, *Z* will resemble *R*. The possibility of a discrepancy between the measured and calculated values of *Z* was expected to appear at higher frequencies, however well below 1 MHz.

Three equivalent circuits were composed to represent the *Z* of the skin: a single resistor (100 Ω) (Figure 5b, circuit A); series connection of the resistor (1 kΩ) and capacitor (22 nF) (Figure 5b, circuit B); parallel connection of the resistor (1 kΩ) and capacitor (22 nF) (Figure 5b, circuit C). General through-hole 0.5 W metal film resistors and ceramic capacitors were utilized and soldered to the breadboard PCB.

Resistors with a value of 10 kΩ represent the resistances of the current-carrying electrodes, while the resistances of the measuring electrodes were considered negligible.

The impedance spectroscope HF2IS together with the HF2TA transimpedance amplifier (Zurich Instruments AG, Zurich, Switzerland) was used to measure the impedance of the composed equivalent circuits with similar parameters as the properties of the skin in the following sections.

The results (measured and calculated curves) in the cases of all three verification circuits are visible in Figure 5a, where *Z* and *F* are shown in a logarithmic scale.

*Z* was calculated according to the equation:(5)$$Z=\sqrt{R^{2}+\left(1/{\left(2\pi fC\right)}^{2}\right)},$$
where *C* is capacitance and *f* is frequency. In the calculated *Z*, the *R* of the current-carrying electrodes was not included to imitate the ideal approach.

Starting at about a frequency of 200 kHz, a deviation of the measured value of *Z* from the calculated value of *Z* appeared. The increase in the value of the measured *Z* at higher frequencies was expected to appear because of the effective parasitic elements in the measurement setup—caused, e.g., by connecting wires of finite length and the input capacitance of the measurement device. Nevertheless, the agreement of the measured values of *Z* with the calculated values of *Z* was obvious in the chosen frequency interval, i.e., the chosen measurement setup is valid. However, at frequencies above 200 kHz, the contribution of the parasitic elements to the measured value of *Z* must be considered. Still, this effect can be expected to apply equally (theoretically) to all the measurements of *Z*, i.e., the results were comparable.

### 4.2. Immittance Measurements of the Skin

The results of the immittance measurements of the dry skin and the skin that was treated by mud and water compresses are represented in this section. To detect the possible effect of the mud on the skin, the measured *Z* and *Y* together with the real and imaginary parts are presented individually.

Reference [20] states that focusing on the real and imaginary parts of immittance when evaluating skin status is justified in certain cases. This applies, e.g., in monitoring the hydration of the skin, where the relevant data are claimed to lie in the measured low-frequency *B* [53]. When applying a wet compress, the skin is hydrated, i.e., a similarity can be recognized. In the current research, dry gold-plated electrodes were used, and the polarization impedance was expected to affect the measured immittance at low frequencies.

#### 4.2.1. The Magnitude and Phase Angle of the Measured Impedance of Skin

The calculated means of the measured *Z* and θ of the skin before and after the application of both compresses can be seen in Figure 6 and Figure 7.

Based on the evaluation of the measurement results, a distinguishable response pattern of θ (but, also *Z*), falling into recognizable intervals in the frequency domain, can be noticed. This is explained by the mechanisms (properties of the skin) that apply at different excitation signal frequencies (dispersions). Based on that phenomenon, the utilized frequency range in the current paper was divided into three frequency intervals and followed accordingly in the following sections:100 Hz …10 kHz;10 kHz …1 MHz;1 MHz …20 MHz.

To illustrate the distribution of individual measurement results, Figure 8 can be evaluated, where the values of σ are plotted. Figure 8 shows the σ in the case of *Z* measurement before the application of the mud/water compress. As in this case, presumably, the measured values of *Z* were the highest, these indicated the distribution of the measurement results of 10 volunteers the best.

To identify the effect of mud and water compresses on dry skin, simultaneously, correlation coefficients (*r*) were calculated for the three cases among the three proposed frequency intervals: before the mud/water compress to after the mud compress (BMWC-AMC), before the mud/water compress to after the mud compress (BMWC-AWC), and after the mud compress to after the water compress (AWC-AWC), denoted as $rZ_{BMWC-AMC}$ and $r\theta_{BMWC-AMC}$, $rZ_{BMWC-AWC}$ and $r\theta_{BMWC-AWC}$, and $rZ_{AMC-AWC}$ and $r\theta_{AMC-AWC}$. The results can be seen in Table 2.

The correlation coefficients indicate an important pair of properties of the linear relationship between two variables (data sets): strength (1) and direction (2). Depending on the amount of sample data (which, in the context of the current manuscript, is reasonable—10 volunteers), conclusions can be drawn, especially in the case of significant differences in the calculated statistical data.

#### 4.2.2. The Conductance and Susceptance of the Measured Admittance of Skin

The calculated means of the measured *G* and *B* of the skin before and after the application of mud and water compresses can be seen in Figure 9a,b, respectively.

The complex plane representation gives an idea of the variation of *G* and *B* and is visible in Figure 10.

The calculated correlation coefficients (*r*) in the cases of *G* and *B* among the three proposed frequency intervals for the three skin conditions (BMWC-AMC, BMWC-AWC, and AWC-AWC), denoted as $rG_{BMWC-AMC}$ and $rG_{BMWC-AMC}$, $rG_{BMWC-AWC}$ and $rG_{BMWC-AWC}$, and $rG_{AMC-AWC}$ and $rG_{AMC-AWC}$, can be seen in Table 3.

#### 4.2.3. The Resistance and Reactance of the Measured Impedance of Skin

As shown already in Figure 4, the content of the measured data that are represented by *Z* and *Y* was expected to depend on the construction of an object under test. Generally, the skin (and more so also the SC [19]) is reported to be a layered structure, which in electrical terms resembles a parallel connection of passive electronic components, so *R* and *X* are expected to incorporate additional information.

The calculated means of the measured *R* and *X* of the skin before and after the application of mud and water compresses can be seen in Figure 11a,b respectively.

The representation of the calculated mean *R* and *X* in the complex plane revealed a noticeable difference between the statuses of the skin before and after the application of mud and water compresses, especially in the low- and medium-frequency interval (Figure 12).

As the data in Figure 12 in the case of the measurements before mud and water compress spread over a significantly wider dispersion of values, minimizing the results of the measurements after the mud and water compress visually, Figure 13 is included. Based on Figure 13, a comparison can be made for measurements after the mud and water compresses.

The calculated correlation coefficients (*r*) in the cases of *R* and *X* among the three proposed frequency intervals for the three skin conditions (BMWC-AMC, BMWC-AWC, and AWC-AWC) denoted as $rR_{BMWC-AMC}$ and $rX_{BMWC-AMC}$, $rR_{BMWC-AWC}$ and $rX_{BMWC-AWC}$, and $rR_{AMC-AWC}$ and $rX_{AMC-AWC}$ can be seen in Table 4.

## 5. Discussion

Impedance analysis has been used in characterizing the efficiency of drug delivery through the skin [29,30] and in evaluating the moisturizing effect of creams in cosmetology [65]. However, the monitoring of the effect of pelotherapy is a little-researched area. Still, in theorizing about the possibility of organic substances and minerals soaking into the skin (which can be expected to happen only by passing the skin barrier), the induced change in electrical properties can be assumed to be low only because of the natural properties of mud.

### 5.1. Evaluation of the Effect of Pelotherapy on the Skin through the Analysis of Magnitude and Phase Angle of the Measured Impedance

The mean of the measured *Z* of the skin (Figure 6) showed a vast difference in the cases of the measurements before and after the application of the mud and water compresses. The individual measurement results presented high dispersion—the highest in the case of the dry skin (Figure 8). This is explained by the status of the SC, the properties of which in dry conditions resemble a dielectric, i.e., *Z* was high. After 30 min in contact with the wet compress, the SC moistened, the conductivity increased, and the impedance decreased rapidly.

This difference diminished with increasing frequency, characterizing well the effect of the capacitive properties of the skin (and living tissues). The difference in the means of the measured *Z* after the application of the mud and water compresses was small, suggesting the approach of evaluating the real and imaginary parts of *Z* and *Y* separately.

At frequencies below 10 kHz, the mean results of θ (Figure 7) demonstrated the domination of the inductive properties (θ < 0°)—the current was lagging voltage. Up to about 2 kHz, the most inductive was the calculated mean θ after the application of the water compress and less inductive before the application of the mud and water compresses. The mean θ after the mud compress application was more towards the properties of dry skin. Starting from 2 kHz, the skin inductive properties decreased after the application of the mud and water compresses faster than before the application of the mud and water compresses.

In the second interval, the dry skin gained resistive properties (θ near 0°), while after the application of the mud compress, these were already becoming capacitive (θ > 0°). In this range, after the application of the water compress, the mean θ was more towards the properties of dry skin (confirmed also by the correlation coefficients in Table 2). Some current was expected to travel still through resistive pathways, while some through the capacitive and inductive pathways. In this middle range, the skin’s conductivity after the mud compress was higher than the skin’s conductivity after the water compress and dry skin—reported already in [3]. However, the second frequency interval was quite wide, and the properties at its lower and higher end may result in a different outcome.

In the third interval, the *Z* of the skin rapidly transformed to capacitive—the very-high-frequency current flowed through the layered structure of the skin regardless of the other properties of the matter.

The easiest way to determine the correlation between the three targeted skin statuses was to compare the correlation coefficients. As expected, the calculated *r* was dissimilar in the cases of the three frequency intervals, confirming the correctness of the approach. The correlations were very or moderately strong for *Z* and θ (based on [66]), but still highly frequency-dependent. However, when comparing the correlation coefficients among *Z* and θ in certain frequency intervals, they were generally similar, with slight differences.

### 5.2. Evaluation of the Effect of Pelotherapy on the Skin through the Analysis of Real and Imaginary Parts of Measured Impedance and Admittance

The means of the measured *G* (Figure 9a), *B* (Figure 9b), *R* (Figure 11a), and *X* (Figure 11b) provided differentiating curves through the whole frequency range. The compliance with previously proposed Frequency Interval Nos. 1–3 can generally be observed.

To evaluate the compliances between the measured means of the real and imaginary parts of *Z* and *Y* in the frequency domain, Table 5 can be referred to. Table 5 relies on the calculated correlation coefficients (Table 2). The table represents the condition of the skin, manifested by the measured values of *G*, *B*, *R*, and *X* in comparison with the same parameters of the untreated skin, i.e., Table 5 denotes that the substance that affects the skin is closer to the untreated condition.

For *G* and *B*, in all frequency intervals, the calculated correlation coefficients (Table 3) indicated a strong relationship between the variables. As expected, the effect of the mud and water compress was not high, but very slight; so, there was the possibility of starting to compare these small differences. However, first of all, it is wise to look at the similar coefficients of *R* and *X*. Surprisingly, a noticeable difference between the calculated correlation coefficients of *R* can be seen in the second frequency interval. Specifically, the correlation between the measured *R* before the application of the mud/water compress and after the mud compress was poor (*r* = 0.166), while it was very strong after the application of the mud compress (*r* = 0.941). Based on Table 4, the correlation coefficient between the measured *R* after the mud compress and after the water compress was even poorer (and negative).

This is a remarkable observation, indicating the domain, in light of all the measured data, where the possible effect of pelotherapy is manifesting. The difference in the calculated correlation coefficients was confirmed by *X* in the cases of the measurements before the application of the mud/water compress and after the mud compress (0.681 vs. 0.956 (Table 4)).

To further study the discovered effect, linear regression graphs were generated, and the coefficients of determination ($R^{2}$) were calculated for *R* in the second frequency interval of all three skin conditions (before the application of the mud/water compress and after the application of the mud and water compresses) (Figure 14).

The linear regression analysis confirmed the outcome—a clear difference in the measured values of *R* after the mud and water compress application in the frequency interval of 10 kHz …1 MHz. Specifically, Figure 14a illustrates practically no fit between the *R* of the skin before the application of the mud/water compress and after the application of the mud compress. Figure 14b visualizes the relatively good fit of the *R* of the skin before the application of the mud/water compress and after the application of the water compress. Figure 14c indicates no correlation between the *R* of the skin after the application of the mud and water compresses, confirming the expectations concerning the effect of mud pack therapy.

The result suggested that the model, which applies in pelotherapy effect monitoring, depicted the skin layers of different properties (and conductivity) in series (as represented in Figure 4a)). While the model in the case of a biological object is complex (because, e.g., sweat glands form direct pathways), approximations can be made. The frequency interval where the finding appeared fell into the β-dispersion, referring to the presence of charging effects and polarization in the interfaces of different skin and tissue layers. Based on [24], in the 10 kHz …1 MHz range, the primary contribution originated from viable skin.

### 5.3. Presence of Effect of Pelotherapy through EBI Analysis

All of this poses a question concerning the evidence of the effect of mixtures of disintegrated mud and peat on the functional condition of the human skin. Based on the research results on a large number of volunteers (86 volunteers aged between 18 and 49 years) by using simple capacitance measurement at relatively high frequencies, presented in [32], such mixtures have been reported to moisten the skin. This effect is claimed to originate from the properties of mud, reported to increase the moisture content in the skin [14].

The fact that water is pulled to higher concentrations of salts can explain the effect of pelotherapy increasing the skin’s electrical conductivity [3]. At the same time, seawater baths are reported to decrease the electrical conductivity of the skin [3]. The question concerning the source of more salts appears—either in the skin or curative mud. However, the effect of pelotherapy is multimodal, and the effect on the skin (and the whole body) is known to comprise a variety of phenomena [34]. The effect of pelotherapy cannot be related only to the salts, but has to be addressed as a whole.

At the low frequencies, the current flowed mostly through the conductive portion of the skin, and at the increasing frequencies, the contribution of capacitive elements started to increase. Fat is a poor electrical conductor; however, its properties are also water-repellent. Accordingly, salts can be expected to affect the skin during pelotherapy—being the main target in the case of EBI measurements. Furthermore, the mud composition (presence of chemical and microbiological substances) may affect its exact influence on the human body [3]. Hence, the role of the origin of the mud (mine) may play a difference in focusing on a certain disease.

The outcome was frequency dependent, i.e., the choice of the excitation signal was critical. By relying on existing knowledge, the object of interest was the viable skin; in this medium, the minerals were dissolving. Based on [24], the lowest frequency was around 100 kHz; at this frequency, the contribution of viable skin in the measured EBI started to dominate. The same conclusion was made based on the performed measurement experiment results. However, in all the cases of the performed measurement experiments, the difference manifested already starting from about 10 kHz, i.e., mechanisms related to pelotherapy revealed already at these frequencies. The maximum frequency of the excitation signal was suggested to be above 5 MHz, where the calculated means started to converge.

The effect of pelotherapy is claimed to be highly individual, depending on the skin type [3]. The changes in the measured EBI of the skin have been related to its barrier functions and the volition of chemical enhancers to permeate (biologically active substances—such as humin peloids) [67]. The barrier function is primarily related to the SC [68]. The fact that there is a poor correlation between the *R* of the mud-pack-treated skin and the *R* of the untreated skin (based on the findings of the current research) implies the possibility that some substances permeate certain skin layers. In a certain frequency interval, *R* changed in the reverse direction to the effect of the tap water (basically the wetting).

It is worth noting the porous pathway theory, which claims that polar uncharged solutes and ions permeate the SC via identical pathways [69]. The evidence of a correlation between skin permeability and EBI was reported in [70]. Based on this, the chemically (or biologically) active compartments, e.g., sulfur anions, can be expected to cause a similar pattern in the measured EBI, i.e., the differentiation by using the EBI measurements was assumed to be highly complex.

The presence of the pelotherapy effect on the measured EBI of skin can be claimed to be indicated based on the presented results. However, the exact mechanism of the pelotherapy cannot be determined purely based on non-invasive EBI measurement. The placing of the measurement data into a more complex skin model is needed. Similar to the complex effect of pelotherapy, the complexity of the skin must be considered when locating the measured data of EBI into the framework of the model. Based on the results of the current research, the effect of mud pack therapy on the skin was evident, revealing the poor correlation to the measured *R* of the skin that was treated with tap water.

The imaginary parts of the measured immittance are another modality of high interest. In the 2nd and 3rd frequency intervals, we fell into the range of β-dispersion. At the higher frequency interval of β-dispersion, the cell membranes and their charging effect became successively prevalent compared to the lower frequency interval of β-dispersion, where the extracellular space dominated in its contribution [20]. Therefore, the effect of minerals became increasingly irrelevant, and the contribution of the dielectric properties of different skin layers started to dominate. As the results showed, a difference also in *X* existed in the frequency interval of 10 kHz …1 MHz and was only moderately strong before the application of the mud/water compress and after the application of the mud compress. The exact mechanism needs further analysis; however, the effect was assumed to be a complex of different mechanisms [68]. The chosen frequency interval is a target for further classification and reduction in future research.

Although the concentrations of chemical substances that soak into the skin (and, eventually, the circulatory system) are small, the effect on *R* in the frequency interval of 10 kHz …1 MHz was surprisingly strong. The assumption was that, in the case of using small-area EBI measurement, the contribution of the skin was the highest, i.e., we must consider the layers of skin, from which the SC was the most-affected by the wetting (by the wet mud and water compresses); dry cells were moistened, and electrical connection to the viable skin was created. Generally, the bottom layer of the skin, the hypodermis, is mainly composed of body fat with blood vessels and nerves. In light of the gained results, the following conclusion can be drawn: in the frequency interval 10 kHz …1 MHz, the main contribution of the measured immittance came from the viable skin, also concluded in [24,71]. The higher end of the selected frequency interval can be contributed by the hypodermis.

## 6. Limitations of the Work

The authors realize that the defined attributes for hiring the volunteers into the study were expected to set limitations on the study and the extraction of the measurement results. However, the main reason for such attributes was to limit the group of subjects based on similar physiological characteristics to enhance the manifesting of a possible effect of pelotherapy. A wider sample size and range of chosen attributes will be a matter of thorough study in the next research phase.

Another major limitation of the current work is that all the volunteers were Caucasian. However, the next, exhaustive study will cover a wider group of volunteers.

Furthermore, one controversy is related to the conditions (environmental, physical, etc.) in which pelotherapy is performed. Mud baths are given traditionally in a specific environment (where a specific room and mud temperature and air humidity are applied), either in a spa or in dedicated locations in the natural environment. We performed the mud pack therapy in the laboratory environment, where there are no spa conditions. However, mud pack therapy is gaining more and more attention in rehabilitation today, applied also in home conditions, providing positive results.

## 7. Conclusions

In this paper, we provided evidence of the effect of pelotherapy on the human skin present in the measured data of EBI. We showed that the effect is likely to be observable in the real and imaginary parts of *Z* (*R* and *X*). The developed and presented approach of monitoring the effect of pelotherapy is novel together with the findings of the correlation between the measured values of the EBI of the skin before and after the applied treatment with the mud. Similar findings have, based on our best knowledge, never been published before.

The finding of the poor correlation between the measured values of *R* before the application of the mud/water compress (dry skin) and after the mud compress and the very strong correlation before the application of the mud/water compress (dry skin) and after the water compress is the highlight based on the performed research. This finding was confirmed by the similar (not equally deep, but still distinctive) discovery in the *X* domain. The frequency interval was a key aspect here: the finding appeared in the frequency interval of 10 kHz …1 MHz, which, based on our results, is suggested for mud pack therapy effect detection.

From the viewpoint of pelotherapy, this finding could open a new modality in monitoring the effect of the performed thermal mud procedure. Pelotherapy, which is a promising rehabilitation method (i.e., cardiac) could gain more scientific proof through standardized measurements—based on bioimpedance analysis. Ultimately, medical-grade devices could emerge, relying on the EBI measurements, to give input to the medical diagnostics to adjust the therapy or assess its effect.

The following assumptions can be made based on the mineral concentrations and skin electrical properties. During mud pack therapy, the salts dissolved into the skin, and throughout, the increased moisture caused the measured *R* to differ significantly from the measured *R* after tap water application in the selected frequency interval (10 kHz …1 MHz). However, the differences in *X* at the same second frequency interval refer to changes in the polarization of cellular membranes, proteins, and other organic macromolecules—possibly related to the specific effect of pelotherapy. However, the exact mechanism is complex and still debatable, as the differences in the imaginary parts of the immittance were revealed concurrently, while the separation of the specific and non-specific effects of pelotherapy needs further research.

Referring to the desire of realizing the body-measuring solution as a portable device, the suitable frequency interval, most affected by the effect of pelotherapy, was identified. Moreover, the possibility of using simple electrical measurements, already in use in pharmacology for detecting the amounts of delivered drugs through the skin, is expected to open a new modality in providing evidence of the effect of pelotherapy and eventually promoting spa procedures.

Further research on the effect of pelotherapy on the skin is planned to gain elaborate measurement results of EBI for comparison and analysis. The number of volunteers will be increased together with an expanded range of requirements. Attention will be turned to the choice of the electrode because of the indistinct effect of the utilized classical tetrapolar electrode configuration—implemented on a rigid PCB. The electrode will be highly flexible and contain hydrogel between the skin and electrode contact surfaces to achieve a good electrical connection to the skin.

As a future target, the recovery of the skin from the effect of pelotherapy will be added to the research plan—-implemented as a time-based measurement of EBI after the removal of the mud. The tape-stripping method will be considered to mechanically disrupt the skin barrier. Related to the interest in the ability of biologically active substances in the form of humin peloids to pass through the skin barrier and their positive effect on blood micro-circulation and the reduction of inflammation, the development of a suitable method will be decided.

## Figures and Tables

**Figure 1 sensors-23-04251-f001:**
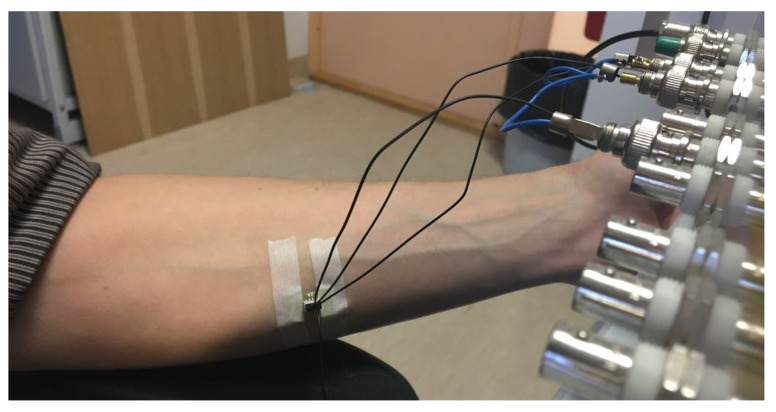
Placement of the custom-designed electrode on the skin for measuring the electrical bioimpedance (EBI).

**Figure 2 sensors-23-04251-f002:**
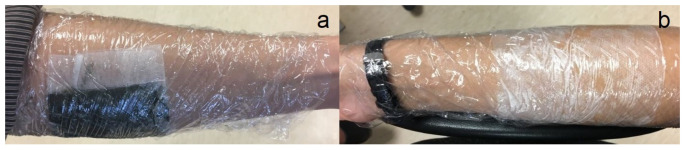
Attachment of mud compress (**a**) and water compress (**b**) on the forearm of a volunteer.

**Figure 3 sensors-23-04251-f003:**
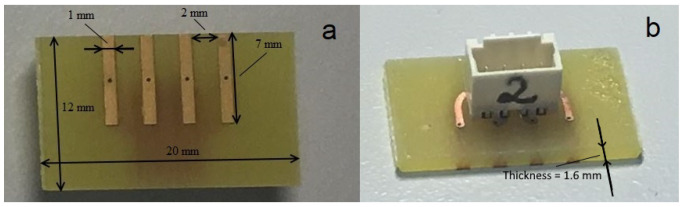
Bottom (electrode) (**a**) and top (connector) (**b**) side of the custom-designed printed circuit board (PCB).

**Figure 4 sensors-23-04251-f004:**
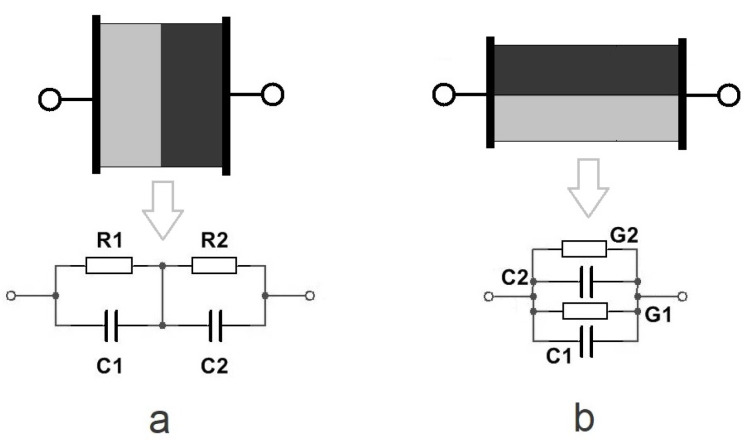
Visual representation of structures with layers of different materials in series (**a**) and in parallel (**b**) with the corresponding equivalent circuits.

**Figure 5 sensors-23-04251-f005:**
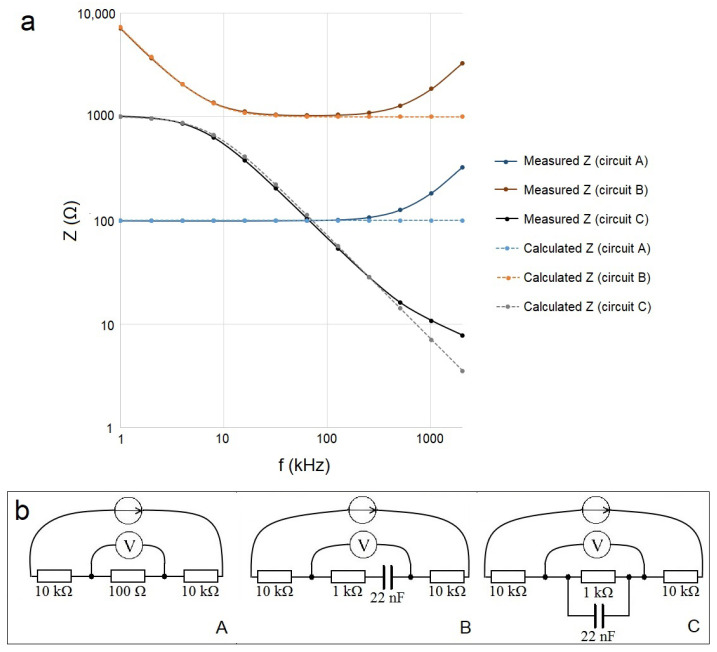
Measured and calculated curves of impedance (*Z*) in the frequency scale (**a**) and the composed equivalent circuits for the verification of the measurement setup (**b**) in the cases of a single resistor (A), series (B); and parallel connection of the resistor and capacitor (C) (where the arrow in the circle means the excitation alternating current source, the letter V in the circle means the voltage measurement, the boxes with the values denote the resistors, and the two bold lines in parallel denote the capacitor).

**Figure 6 sensors-23-04251-f006:**
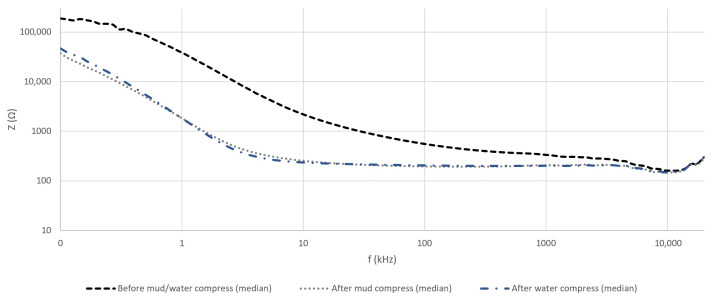
Calculated means of measured *Z* in the cases of all volunteers before and after the application of mud and water compresses in the frequency domain.

**Figure 7 sensors-23-04251-f007:**
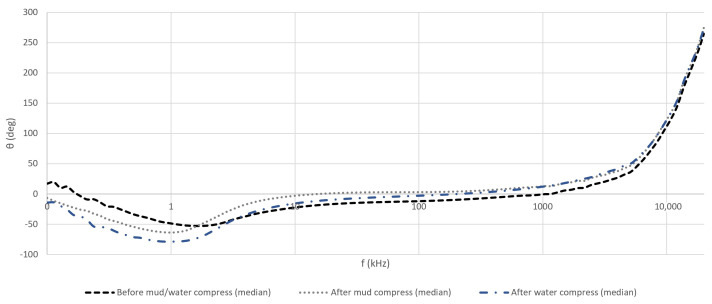
Calculated means of measured phase angle (θ) in the cases of all volunteers before and after the application of mud and water compresses in the frequency domain.

**Figure 8 sensors-23-04251-f008:**
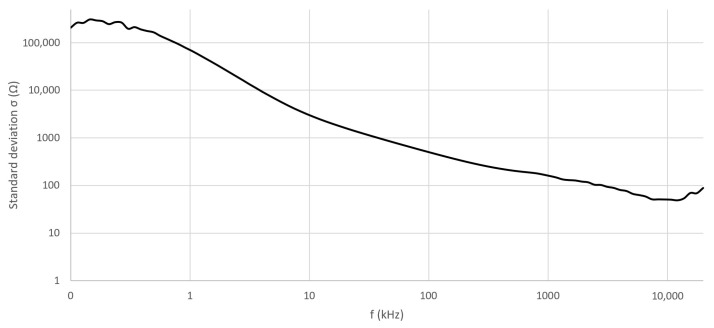
Calculated standard deviations (σ) of measured values of *Z* before the application of mud/water compresses throughout the whole frequency range.

**Figure 9 sensors-23-04251-f009:**
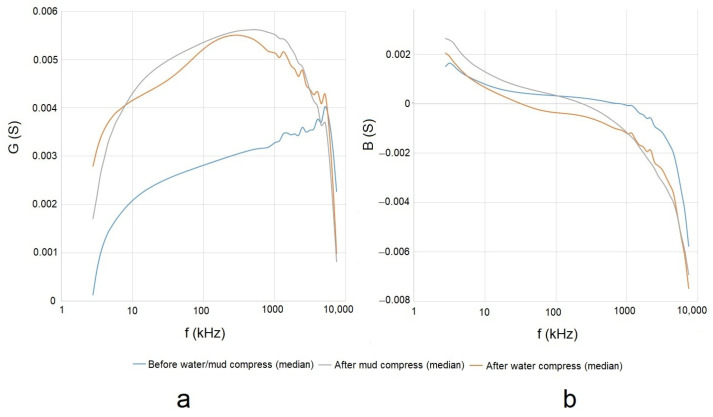
Means of measured conductance (*G*) (**a**) and susceptance (*B*) (**b**) before and after the application of mud and water compresses in the frequency domain.

**Figure 10 sensors-23-04251-f010:**
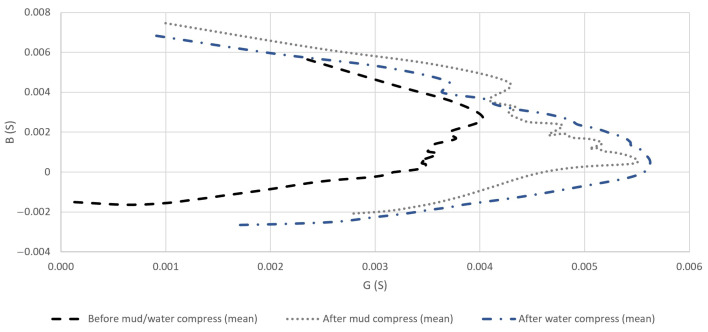
Means of measured admittance (*Y*) (by its real (*G*) and imaginary (*B*) parts) in the cases of the performed measurements before and after the application of mud and water compresses in the complex plane in the full frequency interval.

**Figure 11 sensors-23-04251-f011:**
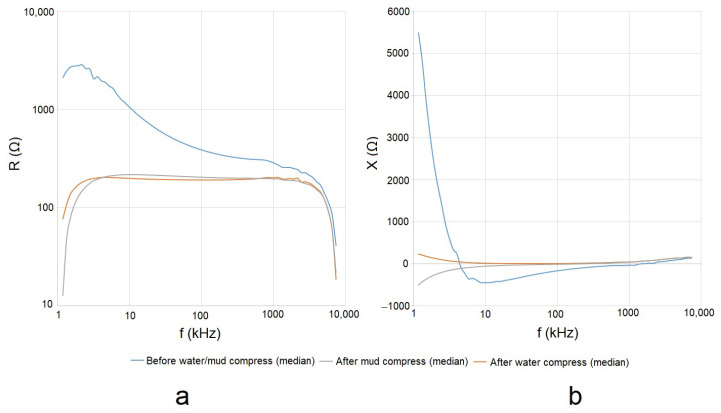
Means of measured real (*R*) (**a**) and imaginary (*X*) (**b**) parts of *Z* before and after the application of mud and water compresses in the frequency domain.

**Figure 12 sensors-23-04251-f012:**
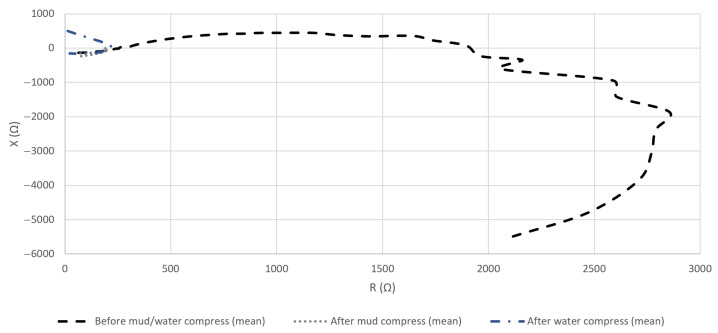
Means of measured *Z* (by its real (*R*) and imaginary (*X*) parts) in the complex plane in the cases of all performed measurements before and after the application of mud and water compresses in the full frequency interval.

**Figure 13 sensors-23-04251-f013:**
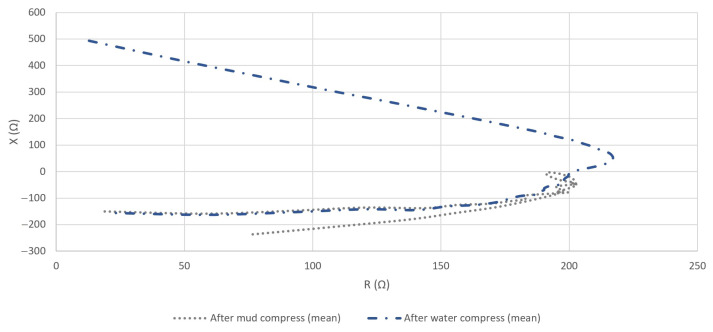
Means of measured *Z* in the case of measurement after the application of mud and water compresses in the complex plane in the full frequency interval.

**Figure 14 sensors-23-04251-f014:**
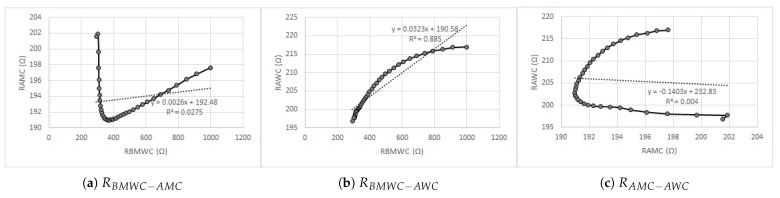
Linear regression graphs for the measured values of *R* in the 10 kHz …1 MHz frequency interval in the cases of before the mud/water compress to after the mud compress ($rR_{BMWC-AMC}$) (**a**), before the mud/water compress to after the water compress ($rR_{BMWC-AWC}$) (**b**), and after the mud compress to after the water compress ($rR_{AMC-AWC}$) (**c**), together with the linear regression equations and calculated coefficients of determination ($R^{2}$).

**Table 1 sensors-23-04251-t001:** The calculated means of the age and physiological data of the volunteers.

Age ± *σ*	Weight ± *σ* (kg)	Height ± *σ* (cm)	BMI ± *σ*
32.5 ± 3.1	75.5 ± 15.6	1.76 ± 0.07	24.19 ± 3.87

**Table 2 sensors-23-04251-t002:** Calculated correlation coefficients (*r*) of the measured values of *Z* and θ for illustrating the strength of a linear relationship between the following conditions: before the mud/water compress to after the mud compress ($rZ_{BMWC-AMC}$ and $r\theta_{BMWC-AMC}$), before the mud/water compress to after the mud compress ($rZ_{BMWC-AWC}$ and $r\theta_{BMWC-AWC}$), and after the mud compress to after the water compress ($rZ_{AMC-AWC}$ and $r\theta_{AMC-AWC}$).

Frequency Interval	${\mathrm{rZ}}_{\mathrm{BMWC}-\mathrm{AMC}}$	$r\theta_{\mathrm{BMWC}-\mathrm{AMC}}$	${\mathrm{rZ}}_{\mathrm{BMWC}-\mathrm{AWC}}$	$r\theta_{\mathrm{BMWC}-\mathrm{AWC}}$	${\mathrm{rZ}}_{\mathrm{AMC}-\mathrm{AWC}}$	$r\theta_{\mathrm{AMC}-\mathrm{AWC}}$
100 Hz–0 kHz	0.914	0.578	0.912	0.668	0.999	0.986
10 kHz–1 MHz	0.954	0.976	0.993	0.998	0.943	0.975
1 MHz–20 MHz	0.795	0.999	0.671	0.999	0.978	0.999
Full	0.926	0.968	0.92	0.962	0.999	0.996

**Table 3 sensors-23-04251-t003:** Calculated correlation coefficients (*r*) of the measured values of *G* and *B* for illustrating the strength of a linear relationship between the following conditions: before the mud/water compress to after the mud compress ($rG_{BMWC-AMC}$ and $rB_{BMWC-AMC}$), before the mud/water compress to after the mud compress ($rG_{BMWC-AWC}$ and $rB_{BMWC-AWC}$), and after the mud compress to after the water compress ($rG_{AMC-AWC}$ and $rB_{AMC-AWC}$).

Frequency Interval	${\mathrm{rG}}_{\mathrm{BMWC}-\mathrm{AMC}}$	${\mathrm{rB}}_{\mathrm{BMWC}-\mathrm{AMC}}$	${\mathrm{rG}}_{\mathrm{BMWC}-\mathrm{AWC}}$	${\mathrm{rB}}_{\mathrm{BMWC}-\mathrm{AWC}}$	${\mathrm{rG}}_{\mathrm{AMC}-\mathrm{AWC}}$	${\mathrm{rB}}_{\mathrm{AMC}-\mathrm{AWC}}$
100 Hz–10 kHz	0.867	0.765	0.965	0.931	0.962	0.934
10 kHz–1 MHz	0.928	0.996	0.98	0.983	0.97	0.979
1 MHz–20 MHz	0.967	0.942	0.953	0.92	0.997	0.997
Full	0.939	0.932	0.938	0.926	0.991	0.984

**Table 4 sensors-23-04251-t004:** Calculated correlation coefficients (*r*) of the measured values of *R* and *X* for illustrating the strength of a linear relationship between the following conditions: before the mud/water compress to after the mud compress ($rR_{BMWC-AMC}$ and $rX_{BMWC-AMC}$), before the mud/water compress to after the mud compress ($rR_{BMWC-AWC}$ and $rX_{BMWC-AWC}$), and after the mud compress to after the water compress ($rR_{AMC-AWC}$ and $rX_{AMC-AWC}$).

Frequency Interval	${\mathrm{rR}}_{\mathrm{BMWC}-\mathrm{AMC}}$	${\mathrm{rX}}_{\mathrm{BMWC}-\mathrm{AMC}}$	${\mathrm{rR}}_{\mathrm{BMWC}-\mathrm{AWC}}$	${\mathrm{rX}}_{\mathrm{BMWC}-\mathrm{AWC}}$	${\mathrm{rR}}_{\mathrm{AMC}-\mathrm{AWC}}$	${\mathrm{rX}}_{\mathrm{AMC}-\mathrm{AWC}}$
100 Hz–10 kHz	0.927	0.816	0.827	0.771	0.88	0.839
10 kHz–1 MHz	0.166	0.681	0.941	0.956	-0.064	0.83
1 MHz–20 MHz	0.941	0.942	0.945	0.92	0.999	0.999
Full	0.939	0.795	0.856	0.648	0.9	0.801

**Table 5 sensors-23-04251-t005:** The application effect of mud and water compresses concerning the measured electrical properties of dry skin, where the value (i.e., mud or water) denotes which substance’s effect on the skin is closer to the untreated condition.

Frequency Interval	G	B	R	X
100 Hz–10 kHz	water	water	mud	mud
10 kHz–1 MHz	water	mud	water	water
1 MHz–20 MHz	mud	mud	converged	mud

## Data Availability

The data presented in this study are available upon request from the corresponding author. The data are not publicly available due to privacy restrictions.

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
