# Peer review of "Electrical Bioimpedance Analysis for Evaluating the Effect of Pelotherapy on the Human Skin: Methodology and Experiments"

_sensors, 2023, doi:10.3390/s23094251_

Round 1
Reviewer 1 Report
The manuscript ”Electrical Bioimpedance Analysis for Evaluating the Effect of Pelotherapy on the Human Skin: Methodology and Experiments” brings original data regarding the effects of pelotherapy, explored by measuring skin electrical impedance. As an improvement suggestion of this article, the Conclusions can be oriented to the clinical correlation of the data and to underline the contribution of electrical changes to biological and medical outcomes.
Reviewer 2 Report
Dear authors,
This is excellent work and a well-presented flow of content. The only concern I have is why this study did not cover a wider group of volunteers.
Reviewer 3 Report
This paper presents a pilot study on the evaluation of the specific effect of pelotherapy on human skin using electrical bioimpedance (EBI) measurements. Pelotherapy is a traditional procedure of applying curative muds on the skin surface, which has been shown to have positive effects on the human body and cure illnesses. The authors aim to develop a methodology and custom-designed electrodes for assessing the passage of chemical and biologically active compounds of curative mud through the human skin by measuring EBI.
Strong Aspects
1. The paper provides a comprehensive background on the subject, including the layers and properties of the human skin, the existing methods of evaluating the effect of pelotherapy, and the application of EBI in other fields.
2. The study was conducted on ten volunteers, following the principles embodied in the Declaration of Helsinki and local statutory requirements, and under the approval of the Tallinn Medical Research Ethics Committee (Estonia).
3. The methodology of the study includes local area mud pack and simultaneous tap water compress application on forearms, in comparison with the measurements of dry skin.
Drawbacks
1. The way the authors have written the impedance information in the line 124,125 and 126 is unclear. What do they mean by the linear conditions?
2. Despite the authors' claim that the electrode pairs have fixed gaps and a width of 1mm, the fabricated electrodes shown in Figure 3 exhibit non-uniform dimensions, with a larger width observed at the bottom side edges. How did the authors account for this deviation from the intended electrode design when measuring and modelling the equivalent circuit?
3. The fabrication method of this electrode should be mentioned in the manuscript. Meantime the thickness of the electrode also should be mentioned in the figure 3.
4. The purpose of a current source, as shown in Figure 5(b), has not been explained in any part of the given context.
5. The clarity of Figures 6 and 7 are inadequate, and they do not meet the standard expected by the journal. The figures require replacement. Consider using distinct colours as all the colours in the current figures appear similar, except for the pink colour.
6. The authors' approach of taking the median value for the almost randomly distributed individual results (figure 6 and 7 )of the Z and θ measurements to determine the parameters in Table 2 appears inadequate. To enhance the reliability of their technique, it is recommended that they consider alternative statistical methods for parameter determination. These methods may include computing the mean and standard deviation of the data or utilizing regression analysis to obtain a more accurate representation of the measurements.
7. Figures 8, 9, and 10 present only the median results, and it would be beneficial to also see the overall distribution of the individual results. As previously mentioned, the authors should consider alternative statistical methods to better represent the data and determine the parameters.
8. The clarity of Figures 11 and 12 are questionable, and the insets lacks any clear description.
9. Although the authors acknowledge that their study's sample size is limited, which could potentially affect the generalizability of their findings, they plan to address this issue in future research by increasing the number of volunteers. Nonetheless, despite the small sample size, significant variations in impedance and angle parameters were observed among the selected volunteers. It is important to investigate the possible reasons behind these variations to understand their implications for the study's overall conclusions.
10. The study was conducted in a laboratory environment, which may not accurately reflect the conditions in which pelotherapy is traditionally performed.
11. The paper suggests that the classical tetrapolar electrode configuration implemented on a rigid printed circuit board may have an indistinct effect on the results.

Reviewer 4 Report
Overall the paper describes an experimental pilot study on electrical skin impedance using the 4-point probes method, comparing three conditions, dry skin, water compress (moisture condition without mud), and a mud compress for testing the specific effects of pelotherapy on skin.
The introduction and background sections give a good glance at information to introduce the topic.
The paper is very extensive, and it is difficult to assess what are the novel and its contributions. It should be reduced and focused on the results and discussion in a clear and easy way.
The topic is of interest, but the presented results are missing fundamental and statistical analyses to compare the different conditions.
Figures 9, 11, and 12 are not clear. What are the axis of the 3d space?
Curves of Z and phase over frequency fig. 6 and fig 7. shows the difference between dry and moisture conditions. The difference over water vs. mud must be statistically analyzed.
Table 3 resembles which conditions were similar to dry skin condition. How is the similarity computed?
The discussion section is generally descriptive of the results, with more emphasis on explaining the possible mechanism is expected. Just section 5.3 is correctly addressing and discussing possible relationships.
The conclusions are poorly sustained by the presented results. The third paragraph is more about limitations and future work than a conclusion.
The effect of the dry electrode (plain gold-plated PCB traces) over the skin cannot be neglected and might be, indeed, part of most of the contribution of the measurements at high frequencies.
The main aspect of improving is how the curves for Z, phase, B, G, and X are compared and differentiated. Please add statistical analyses. Overall there is not a clear difference between peloids application and water humectation only.
Reviewer 5 Report
In my opinion, the subject of the manuscript is interesting, but there are some issues that would need to be addressed before publication. The first one is its extension and detailed information on some topics that are note entirely necessary to understand the work done. In some aspects, it resembles a review rather than the report of the findings, which is actually the matter of interest. I also mention below a series of point comments on some technical issues:
1. Lines 11 and 12: the authors talk about “patterns in EBI waveforms”, which could well be used for applied signals rather than for the spectrum as such.
2. In lines 353 to 356 it is not completely clear the relationship between the choices of gold with the properties of the stratum corneum. The authors ought to give a reference where this is better explained.
3. In equation 4, there seems to be some mistakes as there appears an Rw that ought to be negative in order to represent Rinf in the Cole model. Even more, there is no point in talking about a model when the authors do no parameterize the data.
4. In figures 9, 11 and 12, where data is shown in 3D, one axis is not defined. If it is not needed, the authors could better show them in 2D.
5. In figure 7, phase values are very high before the application of water and mud, which suggest a very poor contact between the electrodes and the skin.
6. Lines 552 to 558: the authors write about inductive properties of the skin and state that, with the use of water and mud compresses those properties increase. That could go in the opposite direction as expected, given that inductive phenomena are expected to be produced by parasitic effects of the cables or by the currents going through the surface of the voltage electrodes and not because of tissue properties. On the other side, this behavior is seen at low frequencies, which could suggest that the sign of the phase is not correctly being interpreted. Therefore, it would be worth examining if, in the intervals of the higher frequencies, the effects are actually capacitive in nature, as it is stated in lines 566 to 568.
Finally, as a minor observation, the use of the word “etc.”, does not sound very professional in scientific communications, and it is used 9 times in the text.
Round 2
Reviewer 3 Report
The authors are commended for addressing previous comments and improving the manuscript. However, some concerns remain.
1. Firstly, the quality of the figures needs improvement, and it is suggested that the authors use vector format for better image quality.
2. Secondly, it is noted that the authors have changed from using median values to mean values in Figure 6 and 7 without explanation, yet both versions look the same. How this is possible? In addition, the use of median values in figures 9 and 11 creates confusion, and the authors are encouraged to explain their reasoning for using both median and mean values.
3. Furthermore, the importance of calculating correlation coefficients in tables 2 and 3 is unclear, especially considering the use of median values in the previous version. The authors are advised to highlight the significance of using correlation coefficients.
4. In figures 10 and 13, it is suggested that the authors use different colors instead of arrows to differentiate overlapping lines for clearer visualization.
5. Finally, the use of linear regression in figure 14 is not well-explained, and the authors are urged to provide more detail on its purpose and how it will be utilized, as the linear curve does not appear to be a good fit for the results.
Reviewer 4 Report
The new version integrates the changes according to the previous comments and helps to support the presented information.
It is highly recommended that the authors provide new data to corroborate the current findings and add other populations to see if it is something dependent on skin characteristics.
Since the idea is to support evidence of the effects of mud therapy and how it acts on a skin level, the relation between the electrical observed effects and the physiological pathways must be stablished.
Also, since this kind of therapy is applied over the whole body, the effect of broader humectation areas must also be analyzed.
What is the interpretation of results for having differences only at high frequency in imaginary components? how can this be interpreted physiologically?
are the results dependent on the type of mud being used?
2.14.0.0 2.14.0.0Author Response
Please see the attachment.

Reviewer 5 Report
After reading the response of the authors to my comments, I still feel that the extensioin of the article is too long for any potential reader. However, if the journal is fine with it, I would not objetc to its publication.
A minor final comment: I was very surprised to read in lines 478-9 that "... For the measurement experiment, an alternating current with an amplitude of 500 mV was applied to the prepared equivalent circuits, and the appeared voltage was measured". Was it 500 mA???
